# BATCH NORMALIZATION EXPLAINED

## ABSTRACT

A critically important, ubiquitous, and yet poorly understood ingredient in modern deep networks (DNs) is batch normalization (BN), which centers and normalizes the feature maps. To date, only limited progress has been made understanding why BN boosts DN learning and inference performance; work has focused exclusively on showing that BN smooths a DN's loss landscape. In this paper, we study BN theoretically from the perspective of function approximation; we exploit the fact that most of today's state-of-the-art DNs are continuous piecewise affine (CPA) splines that fit a predictor to the training data via affine mappings defined over a partition of the input space (the so-called "linear regions"). *We demonstrate that BN is an unsupervised learning technique that – independent of the DN's weights or gradient-based learning – adapts the geometry of a DN's spline partition to match the data.* BN provides a "smart initialization" that boosts the performance of DN learning, because it adapts even a DN initialized with random weights to align its spline partition with the data. We also show that the variation of BN statistics between mini-batches introduces a dropout-like random perturbation to the partition boundaries and hence the decision boundary for classification problems. This per mini-batch perturbation reduces overfitting and improves generalization by increasing the margin between the training samples and the decision boundary.

## 1 INTRODUCTION

Deep learning has made major impacts in a wide range of applications. Mathematically, a deep (neural) network (DN) maps an input vector $x$ to a sequence of $L$ *feature maps* $z_\ell$, $\ell = 1, \ldots, L$ by successively applying the simple nonlinear transformation (termed a DN *layer*)

$$z_{\ell+1} = a\left(W_\ell z_\ell + c_\ell\right), \quad \ell = 0, \ldots, L-1 \tag{1}$$

with $z_0 = x$, $W_\ell$ the weight matrix, $c_\ell$ the bias vector, and $a$ an activation operator that applies a scalar nonlinear activation function $a$ to each element of its vector input. The structure of $W_\ell, c_\ell$ controls the type of layer (e.g., circulant matrix for convolutional layer). For regression tasks, the DN prediction is simply $z_L$, while for classification tasks, $z_L$ is often processed through a softmax operator Goodfellow et al. (2016). The DN parameters $W_\ell, c_\ell$ are learned from a collection of training data samples $\mathcal{X} = \{x_i, i = 1, \ldots, n\}$ (augmented with the corresponding ground-truth labels $y_i$ in supervised settings) by optimizing an objective function (e.g., squared error or cross-entropy). Learning is typically performed via some flavor of stochastic gradient descent (SGD) over randomized mini-batches of training data samples $\mathcal{B} \subset \mathcal{X}$ Goodfellow et al. (2016).

While a host of different DN architectures have been developed over the past several years, modern, high-performing DNs nearly universally employ *batch normalization* (BN) Ioffe & Szegedy (2015) to center and normalize the entries of the feature maps using four additional parameters $\mu_\ell, \sigma_\ell, \beta_\ell, \gamma_\ell$. Define $z_{\ell,k}$ as $k^{\text{th}}$ entry of feature map $z_\ell$ of length $D_\ell$, $w_{\ell,k}$ as the $k^{\text{th}}$ row of the weight matrix $W_\ell$, and $\mu_{\ell,k}, \sigma_{\ell,k}, \beta_{\ell,k}, \gamma_{\ell,k}$ as the $k^{\text{th}}$ entries of the BN parameter vectors $\mu_\ell, \sigma_\ell, \beta_\ell, \gamma_\ell$, respectively. Then we can write the BN-equipped layer $\ell$ mapping extending (1) as

$$z_{\ell+1,k} = a\left(\frac{\langle w_{\ell,k}, z_\ell \rangle - \mu_{\ell,k}}{\sigma_{\ell,k}} \gamma_{\ell,k} + \beta_{\ell,k}\right), k = 1, \ldots, D_\ell. \tag{2}$$

The parameters $\mu_\ell, \sigma_\ell$ are computed as the element-wise mean and standard deviation of $W_\ell z_\ell$ for each mini-batch during training and for the entire training set during testing. The parameters $\beta_\ell, \gamma_\ell$

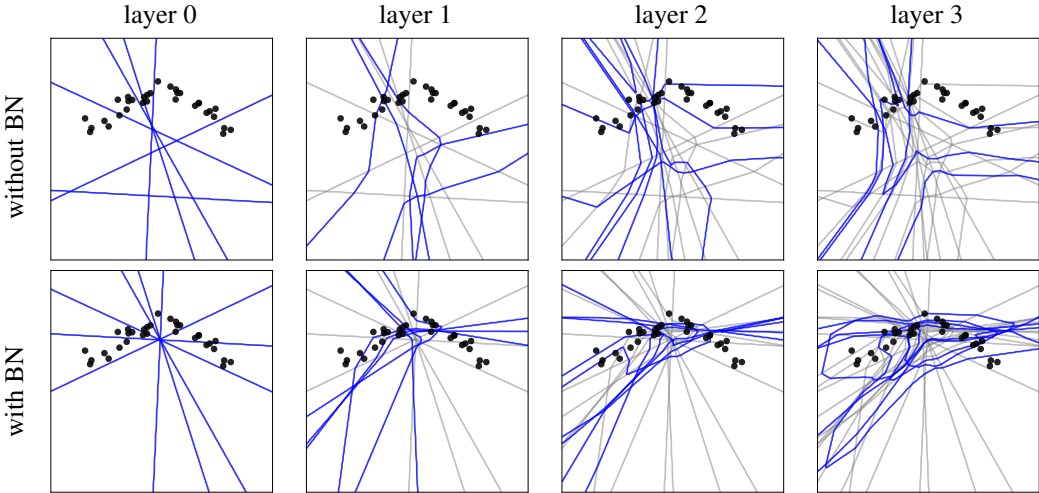

Figure 1: Visualization of the input-space spline partition ("linear regions") of a four-layer DN with 2D input space, 6 units per layer, leaky-ReLU activation function, and random weights $\boldsymbol{W}_\ell$. The training data samples are denoted with black dots. In each plot, blue lines correspond to folded hyperplanes introduced by the units of the corresponding layer, while gray lines correspond to (folded) hyperplanes introduced by previous layers. Top row: Without BN (i.e., using (1)), the folded hyperplanes are spread throughout the input space, resulting in a spline partition that is agnostic to the data. Bottom row: With BN (i.e., using (2)), the folded hyperplanes are drawn towards the data, resulting in an adaptive spline partition that – even with random weights – minimizes the distance between the partition boundaries and the data and thus increases the density of partition regions around the data.

are learned along with $\boldsymbol{W}_\ell$ via SGD.[1] The empirical fact that BN significantly improves both training speed and generalization performance of a DN in a wide range of tasks has made it ubiquitous, as evidenced by the 40,000 citations of the originating paper Ioffe & Szegedy (2015).

Only limited progress has been made to date explaining BN, primarily in the context of optimization. By studying how backpropagation updates the layer weights, LeCun et al. (1998) observed that unnormalized feature maps are constrained to live on a low-dimensional subspace that limits the capacity of gradient-based learning. By slightly altering the BN formula (2), Salimans & Kingma (2016) showed that renormalization via $\boldsymbol{\sigma}_\ell$ smooths the optimization landscape and enables faster training. Similarly, Bjorck et al. (2018); Santurkar et al. (2018); Kohler et al. (2019) confirmed BN's impact on the gradient distribution and optimization landscape through large-scale experiments. Using mean field theory, Yang et al. (2019) characterized the gradient statistics of BN in fully connected feed-forward networks with random weights to show that it regularizes the gradients and improves the optimization landscape conditioning.

One should not take away from the above analyses that BN's only effect is to smooth the optimization loss surface or stabilize gradients. If this were the case, then BN would be redundant in advanced architectures like residual Li et al. (2017) and mollifying networks Gulcehre et al. (2016) that have been proven to have improved optimization landscapes Li et al. (2018); Riedi et al. (2022) and have been coupled with advanced optimization techniques like Adam Kingma & Ba (2014). Quite to the contrary, BN significantly improves the performance of even these advanced networks and techniques.

In this paper, we study BN theoretically from a different perspective that provides new insights into how it boosts DN optimization and inference performance. Our perspective is function approximation; we exploit the fact that most of today's state-of-the-art DNs are *continuous piecewise affine (CPA) splines* that fit a predictor to the training data via affine mappings defined over a partition of the input space (the so-called "linear regions"); see Balestriero & Baraniuk (2021; 2018); Balestriero et al. (2019) and Appendix B for more details.

---

[1]Note that the DN bias $\boldsymbol{c}_\ell$ from (1) has been subsumed into $\boldsymbol{\mu}_\ell$ and $\boldsymbol{\beta}_\ell$.

The key finding of our study is that **BN is an unsupervised learning technique that – independent of the DN's weights or gradient-based learning – adapts the geometry of a DN's spline partition to match the data**. Our three main theoretical contributions are as follows:

- BN adapts the layer/DN input space spline partition to minimize the total least squares (TLS) distance between the spline partition boundaries and the layer/DN inputs, thereby increasing the number of partition regions around the training data and enabling finer approximation (see Figure 1). The BN parameter $\boldsymbol{\mu}_\ell$ translates the boundaries towards the data, while the parameter $\boldsymbol{\sigma}_\ell$ folds the boundaries towards the data (see Sections 2 and 3).
- BN's adaptation of the spline partition provides a "smart initialization" that boosts the performance of DN learning, because it adapts even a DN initialized with random weights $\boldsymbol{W}_\ell$ to align the spline partition to the data (see Section 4).
- BN's statistics vary between mini-batches, which introduces a dropout-like random jitter perturbation to the partition boundaries and hence the decision boundary for classification problems. This jitter reduces overfitting and improves generalization by increasing the margin between the training samples and the decision boundary (see Section 5).

The proofs for our results are provided in the Appendices; the codebase will be released upon completion of the review process to reproduce all the figures and results of the study.

## 2 SINGLE-LAYER ANALYSIS OF BATCH NORMALIZATION

In this section, we investigate how BN impacts one individual DN layer. Our analysis leverages the identification that DN layers using continuous piecewise linear activation functions $a$ in (1) and (2) are *splines* that partition their input space into convex polytopal regions. We show that the BN parameter $\boldsymbol{\mu}_\ell$ translates the regions such that they concentrate around the training data.

### 2.1 BATCH NORMALIZATION DETAILS

The BN parameters $\boldsymbol{\beta}_\ell, \boldsymbol{\gamma}_\ell$, along with the DN weights $\boldsymbol{W}_\ell$, are *learned directly* through the optimization of the DN's objective function (e.g., squared error or cross-entropy) evaluated at the training data samples $\mathcal{X} = \{\boldsymbol{x}_i, i = 1, \dots, n\}$ and labels (if available). Current practice performs the optimization using some flavor of stochastic gradient descent (SGD) over randomized mini-batches of training data samples $\mathcal{B} \subset \mathcal{X}$. Our first new result is that we can set $\boldsymbol{\gamma}_\ell = \mathbf{1}$ and $\boldsymbol{\beta}_\ell = \mathbf{0}$ with no or negligible impact on DN performance for current architectures, training datasets, and tasks. First, we prove in Appendix D that we can set $\boldsymbol{\gamma}_\ell = \mathbf{1}$ both in theory and in practice.

**Proposition 1.** *The BN parameter $\boldsymbol{\gamma}_\ell \neq \mathbf{0}$ does not impact the approximation expressivity of a DN, because its value can be absorbed into $\boldsymbol{W}_{\ell+1}, \boldsymbol{\beta}_\ell$.*

Second, we demonstrate numerically in Appendix D that setting $\boldsymbol{\beta}_\ell = \mathbf{0}$ has negligible impact on DN performance. Henceforth, we will assume for our theoretical analysis that $\boldsymbol{\gamma}_\ell = \mathbf{1}, \boldsymbol{\beta}_\ell = \mathbf{0}$ and will clarify for each experiment whether or not we enforce these constraints.

Let $\mathcal{X}_\ell$ denote the collection of feature maps $\boldsymbol{z}_\ell$ at the input to layer $\ell$ produced by all inputs $\boldsymbol{x}$ in the entire training data set $\mathcal{X}$, and similarly let $\mathcal{B}_\ell$ denote the collection of feature maps $\boldsymbol{z}_\ell$ at the input to layer $\ell$ produced by all inputs $\boldsymbol{x}$ in the mini-batch $\mathcal{B}$.

For each mini-batch $\mathcal{B}$ during training, the BN parameters $\boldsymbol{\mu}_\ell, \boldsymbol{\sigma}_\ell$ are *calculated directly* as the mean and standard deviation of the current mini-batch feature maps $\mathcal{B}_\ell$

$$\boldsymbol{\mu}_\ell \leftarrow \frac{1}{|\mathcal{B}_\ell|} \sum_{\boldsymbol{z}_\ell \in \mathcal{B}_\ell} \boldsymbol{W}_\ell \boldsymbol{z}_\ell, \qquad \boldsymbol{\sigma}_\ell \leftarrow \sqrt{\frac{1}{|\mathcal{B}_\ell|} \sum_{\boldsymbol{z}_\ell \in \mathcal{B}_\ell} \left(\boldsymbol{W}_\ell \boldsymbol{z}_\ell - \boldsymbol{\mu}_\ell\right)^2}, \qquad (3)$$

where the right-hand side square is taken element-wise. After SGD learning is complete, a final fixed "test time" mean $\overline{\boldsymbol{\mu}}_\ell$ and standard deviation $\overline{\boldsymbol{\sigma}}_\ell$ are computed using the above formulae over all of the training data,[2] i.e., with $\mathcal{B}_\ell = \mathcal{X}_\ell$. Note that *no label information enters into the calculation of $\boldsymbol{\mu}_\ell, \boldsymbol{\sigma}_\ell$*.

---

[2] or more commonly as an exponential moving average of the training mini-batch values.

## 2.2 DEEP NETWORK SPLINE PARTITION (ONE LAYER)

We focus on the lionshare of modern DNs that employ continuous piecewise-linear activation functions $a$ in (1) and (2). To streamline our notation, but without loss of generality, we assume that $a$ consists of exactly two linear pieces that connect at the origin, such as the ubiquitous ReLU ($a(u) = \max(0, u)$), leaky-ReLU ($a(u) = \max(\alpha, u), \alpha > 0$), and absolute value ($a(u) = \max(-u, u)$). The extension to more general continuous piecewise-linear activation functions is straightforward Balestriero & Baraniuk (2018; 2021); moreover, the extension to an infinite class of smooth activation functions (including the sigmoid gated learning unit, among others) follows from a simple probabilistic argument Balestriero & Baraniuk (2019). Inserting pooling operators Goodfellow et al. (2016) between layers does not impact our results (see Appendix B).

A DN layer $\ell$ equipped with BN and employing such a piecewise-linear activation function is a *continuous piecewise-affine (CPA) spline operator* defined by a partition $\Omega_\ell$ of the layer's input space $\mathbb{R}^{D_\ell}$ into a collection of convex polytopal regions and a corresponding collection of affine transformations (one for each region) mapping layer inputs $z_\ell$ to layer outputs $z_{\ell+1}$. Here we explain how the partition regions in $\Omega_\ell$ are formed; then in Section 2.3 we begin our investigation of how these regions are transformed by BN.

Define the *pre-activation* of layer $\ell$ by $h_\ell$ such that the layer output $z_{\ell+1} = a(h_\ell)$; from (2) its $k^{\text{th}}$ entry is given by

$$h_{\ell,k} = \frac{\langle w_{\ell,k}, z_\ell \rangle - \mu_{\ell,k}}{\sigma_{\ell,k}}. \tag{4}$$

Note from (3) that $\sigma_{\ell,k} > 0$ as long as $\|w_{\ell,k}\|_2^2 > 0$ and as long as not all inputs are orthogonal to $w_{\ell,k}$. With typical CPA nonlinearities $a$, the $k^{\text{th}}$ feature map output $z_{\ell+1,k} = a(h_{\ell,k})$ is linear in $h_{\ell,k}$ for all inputs with same sign. The separation between those two linear regimes is formed by the collection of layer inputs $z_\ell$ that produce pre-activations with $h_{\ell,k} = 0$, hence lie on the $D_\ell - 1$ dimensional hyperplane

$$\mathcal{H}_{\ell,k} = \left\{ z_\ell \in \mathbb{R}^{D_\ell} : h_{\ell,k} = 0 \right\} = \left\{ z_\ell \in \mathbb{R}^{D_\ell} : \langle w_{\ell,k}, z_\ell \rangle = \mu_{\ell,k} \right\}. \tag{5}$$

Note that $\mathcal{H}_{\ell,k}$ is independent of the value of $\sigma_{\ell,k}$. The boundary $\partial\Omega_\ell$ of the layer's input space partition $\Omega_\ell$ is obtained simply by combining all of the $\mathcal{H}_{\ell,k}$ into the *hyperplane arrangement* Zaslavsky (1975)

$$\partial\Omega_\ell = \cup_{k=1}^{D_\ell} \mathcal{H}_{\ell,k}. \tag{6}$$

For additional results on the DN spline partition, see Montufar et al. (2014); Raghu et al. (2017); Serra et al. (2018); Balestriero et al. (2019); the only property of interest here is that, for all inputs lying in the same region $\omega \in \Omega_\ell$, the layer mapping is a simple affine transformation $z_\ell = \sum_{\omega \in \Omega}(A_\ell(\omega) z_{\ell-1} + b_\ell(\omega)) \mathbb{1}_{\{z_{\ell-1} \in \omega\}}$ (see Appendix B).

## 2.3 BATCH NORMALIZATION PARAMETER $\mu$ TRANSLATES THE SPLINE PARTITION BOUNDARIES TOWARDS THE TRAINING DATA (PART 1)

With the above background in place, we now demonstrate that the BN parameter $\mu_\ell$ impacts the spline partition $\Omega_\ell$ of the input space of DN layer $\ell$ by *translating its boundaries $\partial\Omega_\ell$ towards the current mini-batch training data $\mathcal{X}_\ell$.*

We begin with some definitions. The Euclidean distance from a point $v$ in layer $\ell$'s input space to the layer's $k^{\text{th}}$ hyperplane $\mathcal{H}_{\ell,k}$ is easily calculated as (e.g., Eq. 1 in Amaldi & Coniglio (2013))

$$d(v, \mathcal{H}_{\ell,k}) = \frac{|\langle w_{\ell,k}, v \rangle - \mu_{\ell,k}|}{\|w_{\ell,k}\|_2} \tag{7}$$

as long as $\|w_{\ell,k}\| > 0$. Then, the average squared distance between $\mathcal{H}_{\ell,k}$ and a collection of points $\mathcal{V}$ in layer $\ell$'s input space is given by

$$\mathcal{L}_k(\mu_{\ell,k}, \mathcal{V}) = \frac{1}{|\mathcal{V}|} \sum_{v \in \mathcal{V}} d(v, \mathcal{H}_{\ell,k})^2 = \frac{\sigma_{\ell,k}^2}{\|w_{\ell,k}\|_2^2}, \tag{8}$$

where we have made explicit the dependency of $\mathcal{L}_k$ on $\mu_{\ell,k}$ through $\mathcal{H}_{\ell,k}$. Going one step further, the *total least squares (TLS) distance* (Samuelson, 1942; Golub & Van Loan, 1980) between a collection of points $\mathcal{V}$ in layer $\ell$'s input space and layer $\ell$'s partition $\Omega_\ell$ is given by

$$\mathcal{L}(\boldsymbol{\mu}_\ell, \mathcal{V}) = \sum_{k=1}^{D_\ell} \mathcal{L}_k(\mu_{\ell,k}, \mathcal{V}). \tag{9}$$

In Appendix E.1, we prove that the BN parameter $\boldsymbol{\mu}_\ell$ as computed in (3) is the unique solution of the strictly convex optimization problem of minimizing the average TLS distance (9) between the training data and layer $\ell$'s hyperplanes $\mathcal{H}_{\ell,k}$ and hence spline partition region boundaries $\partial\Omega_\ell$.

**Theorem 1.** *Consider layer $\ell$ of a DN as described in (2) and a mini-batch of layer inputs $\mathcal{B}_\ell \subset \mathcal{X}_\ell$. Then $\boldsymbol{\mu}_\ell$ in (3) is the unique minimizer of $\mathcal{L}(\boldsymbol{\mu}_\ell, \mathcal{B}_\ell)$, and $\overline{\boldsymbol{\mu}}_\ell$ is the unique minimizer of $\mathcal{L}(\boldsymbol{\mu}_\ell, \mathcal{X}_\ell)$.*

In words, at each layer $\ell$ of a DN, BN explicitly adapts the input-space partition $\Omega_\ell$ by using $\boldsymbol{\mu}_\ell$ to translate its boundaries $\mathcal{H}_{\ell,1}, \mathcal{H}_{\ell,2}, \ldots$ to minimize the TLS distance to the training data. Figure 1 demonstrates empirically in two dimensions how this translation focuses the layer's spline partition on the data. Moreover, this translation takes on a very special form. We prove in Appendix E.2 that BN transforms the spline partition boundary $\partial\Omega_\ell$ into a *central hyperplane arrangement* Stanley et al. (2004).

It is worth noting that the above results do not involve any data label information, and so – at least as far as $\boldsymbol{\mu}$ and $\boldsymbol{\sigma}$ are concerned – BN can be interpreted as an *unsupervised* learning technique.

## 3 MULTIPLE LAYER ANALYSIS OF BATCH NORMALIZATION

We now extend the single-layer analysis of the previous section to the composition of two or more DN layers. We begin by showing that the layers' $\boldsymbol{\mu}_\ell$ continue to translate the hyperplanes that comprise the spline partition boundary such that they concentrate around the training data. We then show that the layers' $\boldsymbol{\sigma}_\ell$ fold those same hyperplanes with the same goal.

### 3.1 DEEP NETWORK SPLINE PARTITION (MULTIPLE LAYERS)

Taking advantage of the fact that a composition of multiple CPA splines is itself a CPA spline, we now extend the results from Section 2.2 regarding one DN layer to the composition of layers $1, \ldots, \ell, \ell > 1$ that maps the DN input $\boldsymbol{x}$ to the feature map $\boldsymbol{z}_{\ell+1}$.[3] The denote partition of this mapping by $\Omega_{|\ell}$, where we introduce the shorthand $|\ell$ to denote the mapping through layers $1, \ldots, \ell$. Appendix B provides closed-form formulas for the per-region affine mappings.

As in Section 2.2, we are primarily interested in the boundary $\partial\Omega_{|\ell}$ of the spline partition $\Omega_{|\ell}$. Recall that the boundary of the spline partition of a single layer was easily found in (4)–(6) as the rearrangement of the hyperplanes formed where the layer's pre-activation equals zero. With multiple layers, the situation is almost the same as in (5)

$$\partial\Omega_{|\ell} = \bigcup_{j=1}^{\ell} \bigcup_{k=1}^{D_j} \left\{ \boldsymbol{x} \in \mathbb{R}^{D_1} : h_{j,k} = 0 \right\}; \tag{10}$$

further details are provided in Appendix B. The salient result of interest to us is that $\partial\Omega_{|\ell}$ is constructed from the hyperplanes $\mathcal{H}_{j,k}$ pulled back through the preceding layer(s). This process *folds* those hyperplanes (toy depiction given in Figure 8) based on the preceding layers' partitions such that the folded $\mathcal{H}_{j,k}$ consist of a collection of *facets*

$$\mathcal{F}_{j,k,\omega} = \left\{ \boldsymbol{x} \in \omega : h_{j,k} = 0 \right\} = \left\{ \boldsymbol{x} \in \omega : \langle \boldsymbol{w}_{j,k}, \boldsymbol{z}_j \rangle = \mu_{j,k} \right\}, \quad \omega \in \Omega_{|j}, \tag{11}$$

which simplifies (10) to $\partial\Omega_{|\ell} = \bigcup_{j=1}^{\ell} \bigcup_{k=1}^{D_j} \mathcal{F}_{j,k}$, where $\mathcal{F}_{\ell,k} \triangleq \bigcup_{\omega \in \Omega_{|j}} \mathcal{F}_{j,k,\omega}$.

---

[3]Our analysis applies to any composition of $\ell$ DN layers; we focus on the first $\ell$ layers only for concreteness.

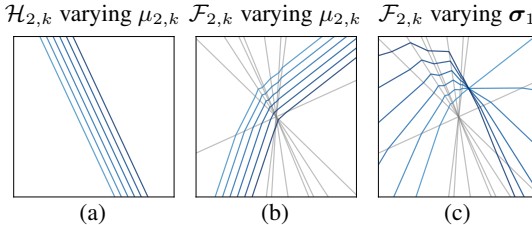

$\mathcal{H}_{2,k}$ varying $\mu_{2,k}$  $\mathcal{F}_{2,k}$ varying $\mu_{2,k}$  $\mathcal{F}_{2,k}$ varying $\boldsymbol{\sigma}_1$

(a)    (b)    (c)

Figure 2: Translation and folding effected by the BN parameters $\boldsymbol{\mu}_\ell, \boldsymbol{\sigma}_\ell$ on a two-layer DN with 8 units per layer, 2D input space, and random weights. (a) Varying $\mu_{2,k}$ translates the layer 2, unit $k$ hyperplane $\mathcal{H}_{2,k}$ (recall (5)) viewed in a 2D slice of its own input space. (b) Translation of that same hyperplane but now viewed in the DN input space, where it becomes the folded hyperplane $\mathcal{F}_{2,k}$ (recall (11)). (c) Fixing $\mu_{2,k}$ and varying $\boldsymbol{\sigma}_1$ folds the next layer(s) hyperplanes, viewed in the DN input space.

## 3.2 BATCH NORMALIZATION PARAMETER $\boldsymbol{\mu}$ TRANSLATES THE SPLINE PARTITION BOUNDARIES TOWARDS THE TRAINING DATA (PART 2)

In the one-layer case, we saw in Theorem 1 that BN independently translates each DN layer's hyperplanes $\mathcal{H}_{\ell,k}$ towards the training data to minimize the TLS distance. In the multilayer case, as we have just seen, those hyperplanes $\mathcal{H}_{\ell,k}$ become folded hyperplanes $\mathcal{F}_{\ell,k}$ (recall (10)).

We now demonstrate that the BN parameter $\boldsymbol{\mu}_\ell$ translates the folded hyperplanes $\mathcal{F}_{\ell,k}$ – and thus adapts $\Omega_{|\ell}$ – towards the input-space training data $\mathcal{X}$. To this end, denote the squared distance from a point $\boldsymbol{x}$ in the DN input space to the folded hyperplane $\mathcal{F}_{\ell,k}$ by

$$d(\boldsymbol{x}, \mathcal{F}_{\ell,k}) = \min_{\boldsymbol{x}' \in \mathcal{F}_{\ell,k}} \|\boldsymbol{x} - \boldsymbol{x}'\|^2. \tag{12}$$

**Theorem 2.** *Consider layer $\ell > 1$ of a layer as described in (2) with fixed weight matrices and BN parameters from layers 1 through $\ell - 1$ and fixed weights $\mathbf{W}_\ell$. Let $\boldsymbol{\mu}_\ell$ and $\boldsymbol{\mu}'_\ell$ yield the hyperplanes $\mathcal{H}_{\ell,k}$ and $\mathcal{H}'_{\ell,k}$ and their corresponding folded hyperplanes $\mathcal{F}_{\ell,k}$ and $\mathcal{F}'_{\ell,k}$. Then we have that $d(\boldsymbol{z}_\ell(\boldsymbol{x}), \mathcal{H}_{\ell,k}) < d(\boldsymbol{z}_\ell(\boldsymbol{x}), \mathcal{H}'_{\ell,k}) \implies d(\boldsymbol{x}, \mathcal{F}_{\ell,k}) < d(\boldsymbol{x}, \mathcal{F}'_{\ell,k}).$*

In words, translating a hyperplane closer to $\boldsymbol{z}_\ell$ in layer $\ell$'s input space moves the corresponding folded hyperplane closer to the DN input $\boldsymbol{x}$ that produced $\boldsymbol{z}_\ell$, which is of particular interest for inputs $\boldsymbol{x}_i$ from the training data $\mathcal{X}$. We also have the following corollary.

**Corollary 1.** *Consider layer $\ell > 1$ of a trained BN-equipped DN as described in Theorem 2. Then $\boldsymbol{z}_\ell(\boldsymbol{x})$ lies on hyperplane $\mathcal{H}_{\ell,k}$ for some $k$ if and only $\boldsymbol{x}$ lies on the corresponding folded hyperplane $\mathcal{F}_{\ell,k}$ in the DN input space; that is, $d(\boldsymbol{z}_{\ell-1}(\boldsymbol{x}), \mathcal{H}_{\ell,k}) = 0 \iff d(\boldsymbol{x}, \mathcal{F}_{\ell,k}) = 0.$*

Figure 2(b) illustrates empirically how $\boldsymbol{\mu}_2$ translates the folded hyperplanes $\mathcal{F}_{2,k}$ realized by the second layer of a toy DN. The impact of the BN parameters $\boldsymbol{\sigma}$, is also crucial albeit not as crucial as $\boldsymbol{\mu}$. For completeness, we study how this parameter translates and folds adjacent facets composing the folded hyperplanes $\mathcal{F}_{\ell,k}$ in Figure 2 and Appendix C.

## 3.3 BATCH-NORMALIZATION INCREASES THE DENSITY OF PARTITION REGIONS AROUND THE TRAINING DATA

We now extend the toy DN numerical experiments reported in Figures 1 to more realistic DN architectures and higher-dimensional settings. We focus on three settings all involving random weights $\mathbf{W}_\ell$: (i) zero bias $\boldsymbol{c}_\ell = 0$ in (1), (ii) random bias $\boldsymbol{c}_\ell$ in (1), and (iii) BN in (2).

**2D Toy Dataset.** We continue visualizing the effect of BN in a 2D input space by reproducing the experiment of Figure 1 but with a more realistic DN with 11 layers of width 1024 and training data consisting of 50 samples from a star shape in 2D (see the leftmost plots in Figure 3). For each of the above three settings, Figure 3 visualizes in the 2D input space the *concentration* of the contribution to the partition boundary (recall (11)) from three specific layers ($j = 1, 7, 11$). The concentration at each point in the input space corresponds to the number of folded hyperplane facets $\mathcal{F}_{j,k,\omega}$ passing through an $\epsilon$-ball centered on that point. This concentration calculation can be performed efficiently via the technique of Harman & Lacko (2010) that is analyzed in Voelker et al. (2017). Three conclusions are evident from the figure: (i) BN clearly focuses the spline partition on the data; (ii) random bias creates a random partition that is diffused over the entire input space; (iii) zero bias creates a (more or less) central hyperplane arrangement that does not focus on the data.

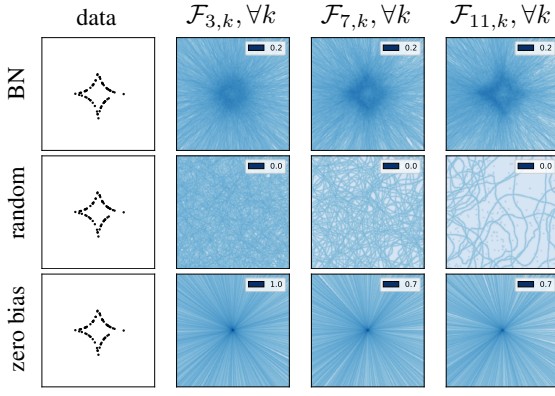

Figure 3: Visualization in the 2D input space of the contribution $\partial\Omega^1_j$ to the spline partition boundary $\Omega^1$ from layers $j = 1, 7, 11$ of an 11-layer DN of width 1024. The training data set $\mathcal{X}$ consists of 50 samples from a star-shaped distribution (left plots). We plot the concentration of the folded hyperplane facets in an $\epsilon$-ball around each 2D input space point for the three initialization settings described in the text: zero bias, random bias, and BN. Darker color indicates more partition boundaries crossing through that location. Each plot is normalized by the maximum concentration attained, the value of which is noted.

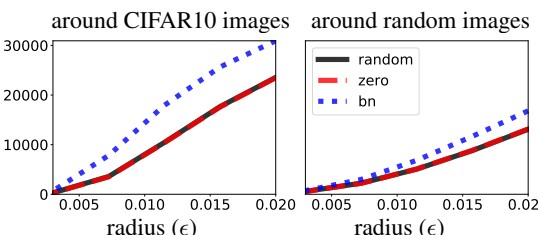

Figure 4: Average concentration of the spline partition boundary $\Omega^1$ of Resnet9 DN in $\epsilon$-balls around CIFAR images and around random images. The vertical axis is the average number of folded hyperplane facets in $\Omega^1$ that pass through an epsilon ball centered on each kind of image. As in Figure 3, we see that – in contrast to zero and random bias initialization – BN aligns the spline partition with the data.

**CIFAR Images.** We now consider a high-dimensional input space (CIFAR images) with a Resnet9 architecture (random weights). Since we cannot visualize the spline partition in the 3072D input space, we present a summary of the same concentration analysis carried out in Figure 3 for the partition boundary by measuring the number of folded hyperplanes passing through an $\epsilon$-ball centered around 100 randomly sampled training images (BN statistics are obtained from the full training set). We report these observation in Figure 4 (left) and again clearly see that – in contrast to zero and random bias initialization – BN focuses the spline partition to lie close to the data points. To quantify the concentration of regions away from the training data, we repeat the same measurement but for 100 iid random Gaussian images that are scaled to the same mean and variance as the CIFAR images. For this case, we observe in Figure 4 (right) that BN only focused the partition around the CIFAR images and not around the random ones. This is in concurrence with the low-dimensional experiment in Figure 3.

## 4 BENEFIT ONE: BATCH NORMALIZATION IS A SMART INITIALIZATION

Up to this point, our analysis has revealed that BN effects a task-independent, unsupervised learning that aligns a DN's spline partition with the training data independent of any labels. We now demonstrate that BN's utility extends to supervised learning tasks like regression and classification that feature both data and labels. In particular, BN can be viewed as providing a "smart initialization" that expedites SGD-based supervised learning. Our results augment recent empirical work on the importance of initialization on DN performance from the perspective of slope variance constraints Mishkin & Matas (2015); Xie et al. (2017), singular value constraints Jia et al. (2017), and orthogonality constraints Saxe et al. (2013); Bansal et al. (2018).

First, BN translates and folds the hyperplanes and facets that construct the DN's spline partition $\Omega$ so that they are closer to the data. This creates more granular partition regions around the training data, which enables better approximation of complicated regression functions DeVore & Lorentz (1993) and richer classification boundaries Balestriero et al. (2019); Chen et al. (2021). For binary classification, for example, this result follows from the fact that the decision boundary created by an $L$-layer DN is precisely the folded hyperplane $\mathcal{F}_{L,1}$ generated by the final layer. We also prove in Proposition 3 in Appendix E.4 that BN primes SGD learning so that the decision boundary passes through the data in every mini-batch.

We now empirically demonstrate the benefits of BN's smart initialization for learning by a comparing the same three initialization strategies from Section 3.3 in a classification problem using a

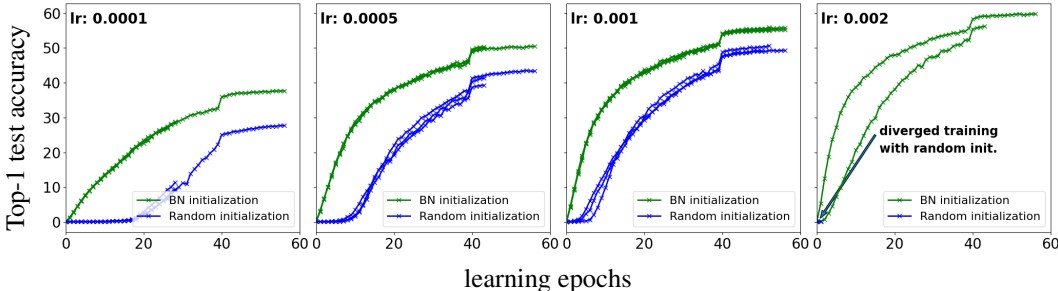

Figure 5: Image classification using a Resnet50 on Imagenet. Standard data augmentation was used during SGD training but no BN was employed. Instead, the DN was initialized with random weights (blue) or with random weights and warmed-up BN statistics () across the entire training set (recall Figure 3 and (3)). Each **column** corresponds to different learning rate used by SGD and multiple runs are produced for the right cases. BN's "smart initialization" reaches a higher-performing solution faster because it provides SGD with a spline partition that is already adapted to the training dataset. This findings provides a novel and complementary understanding of BN that was previously studied solely when continuously employed during training as a mean to better condition a DN's loss landscape. We also show in Figure 5 in the Appendix a similar observation on CIFAR100.

ResNet9 with random weights. In each experiment, the weights and biases are initialized according to one of the three strategies (zero bias, random bias, and BN over the entire training set), and then standard SGD is performed over mini-batches. Importantly, in the BN case, the BN parameters $\boldsymbol{\mu}_\ell, \boldsymbol{\beta}_\ell$ computed for the initialization are frozen for all SGD iterations; this will showcase BN's role as a smart initialization. As we see from Figure 5, SGD with BN initialization converges faster and to an higher-performing classifier. Since BN is only used as an initializer here, the performance gain can be attributed to the better positioning of the ResNet's initial spline partition $\Omega$.

This result showcases the importance of DN initialization and how a good initialization alone plays a crucial role in performance, as has also been empirically studied with slope variance constraints Mishkin & Matas (2015); Xie et al. (2017), singular values constraints Jia et al. (2017) or with orthogonal constraints Saxe et al. (2013); Bansal et al. (2018).

## 5 BENEFIT TWO: BATCH NORMALIZATION INCREASES MARGINS

The BN parameters $\boldsymbol{\mu}_\ell, \boldsymbol{\sigma}_\ell$ are re-calculated for each mini-batch $\mathcal{B}_\ell$ and hence can be interpreted as stochastic estimators of the mean and standard deviation of the complete training data set $\mathcal{X}_\ell$. A direct calculation Von Mises (2014) yields the following result.

**Proposition 2.** *Consider layer $\ell > 1$ of a BN-equipped DN as described in (2). Assume that the layer input $\boldsymbol{z}_\ell$ follows an arbitrary iid distribution with zero mean and diagonal covariance matrix $diag(\boldsymbol{m}\rho)$. Then we have that*

$$\mathrm{var}(\mu_{\ell,k}) = \frac{\langle \boldsymbol{w}_{\ell,k}^2, \boldsymbol{m}\rho \rangle}{|\mathcal{B}_\ell|} \leq \frac{\|\boldsymbol{w}_{\ell,k}^2\| \, \|\boldsymbol{m}\rho\|}{|\mathcal{B}_\ell|}, \quad \mathrm{var}(\sigma_{\ell,k}^2) = \frac{1}{|\mathcal{B}_\ell|} \left( \phi_{\ell,k}^4 - \frac{\langle \boldsymbol{w}_{\ell,k}^2, \rho \rangle^2 (|\mathcal{B}_\ell| - 3)}{|\mathcal{B}_\ell| - 1} \right). \quad (13)$$

*with $\phi_{\ell,k}^4$ the fourth-order central moment of $\langle \boldsymbol{w}_{\ell,k}, \boldsymbol{z}_\ell \rangle$ and $\boldsymbol{w}_{\ell,k}^2$ the coordinate-wise square.*

Consequently, during learning, BN's centering and scaling introduces both additive and multiplicative noise to the feature maps whose variance increases as the mini-batch size $|\mathcal{B}_\ell|$ decreases. This noise becomes detrimental for small mini-batches, which has been empirically observed in Ioffe (2017). We illustrate this result in Figure 6, where we depict the DN decision boundary realizations from different mini-batches. We also provide in Figure 7 the empirical, analytical, and controlled parameter distributions of BN applied on a Gaussian input with varying mean and variance.

Figure 6 suggests the interpretation that BN noise induces "jitter" in the DN decision boundary. Small amounts of jitter can be beneficial, since it forces the DN to learn a representation with an increased margin around the decision boundary. Large amounts of jitter can be detrimental, since the increased margin around the decision boundary might be too large for the classification task. This jitter noise is reminiscent of dropout and other techniques that artificially add noise in the DN

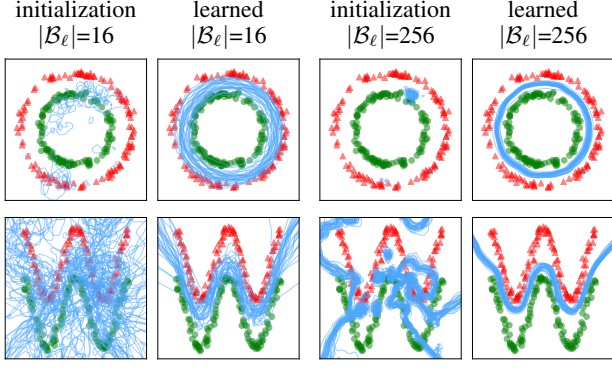

| initialization $|\mathcal{B}_\ell|=16$ | learned $|\mathcal{B}_\ell|=16$ | initialization $|\mathcal{B}_\ell|=256$ | learned $|\mathcal{B}_\ell|=256$ |

Figure 6: Realizations of the classification decision boundary on a toy 2D binary classification (red vs green) problem obtained solely by sampling different mini-batches and thus observing different realizations of $\boldsymbol{\mu}_\ell, \boldsymbol{\sigma}_\ell$ (recall (3)) at initialization and after training. Each mini-batch produces a different decision boundary depicted in blue. For two different mini-batch sizes $|\mathcal{B}_\ell| = 16, 256$, we change the variance as per Proposition 2. Larger batch sizes clearly produce smaller variability in the decision boundary both at initialization and after training; $\boldsymbol{\mu}_\ell, \boldsymbol{\sigma}_\ell$ distributions are provided in Figure 7.

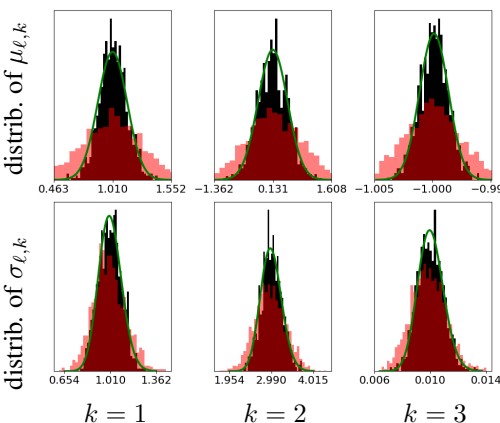

Figure 7: Distributions of $\mu_{\ell,k}$ (top row) and $\sigma_{\ell,k}$ (bottom row) for three units in a DN layer with $\boldsymbol{W}_\ell \boldsymbol{z}_{\ell-1} \sim \mathcal{N}([1, 0, -1], \mathrm{diag}([1, 3, 0.1]))$ ($\ell$ is not relevant in this context). The empirical distributions in **black** are obtained by repeatedly sampling mini-batches of size 64 from a training set of size 1000. The analytical distributions in **green** are from (13). The empirical distributions in **red** are of the noise-controlled BN, where we increase the variances of the BN parameters to match the variances that would result from a virtual mini-batch size (32) that is smaller than the actual size (64), hence producing more regularizing jitter perturbations. The different realizations of $\boldsymbol{\mu}_\ell$ and $\boldsymbol{\sigma}_\ell$ for each mini-batch affect the geometry of the DN partition and decision boundary (see Figure 2) of the current mini-batch.

input and/or feature maps to improve generalization performance Srivastava (2013); Pham et al. (2014); Molchanov et al. (2017); Wang et al. (2018). We focus on the effect of BN jitter on DNs for classification problems here, but jitter will also improve DN performance on regression problems.

To further demonstrate that jitter noise increases the margin between the learned decision boundary and the training set (and hence improves generalization), we conducted an experiment where we fed additional Gaussian additive noise and Chi-square multiplicative noise to the layer pre-activations of a DN to increase the variances of $\boldsymbol{\mu}_\ell$ and $\boldsymbol{\sigma}_\ell$ as desired (as in Figure 7), in addition to the BN-induced noise. For a Resnet9, we observed that increasing these variances about $15\%$ increased the classification accuracies (averaged over 5 runs) from $93.34\%$ to $93.68\%$ (CIFAR10), from $72.22\%$ to $72.74\%$ (CIFAR100) and from $96.16\%$ to $96.41\%$ (SVHN). Note that this performance boost comes in addition to that obtained from BN's smart initialization (recall Section 4).

## 6 CONCLUSIONS

In this paper, our theoretical analysis of BN shed light and explained two novel crucial ways on how BN boosts DNs performances. First, BN provides a "smart initialization" that solve a total least squares (TLS) optimization to adapt the DN input space spline partition to the data to improve learning and ultimately function approximation. Second, for classification applications, BN introduces a random jitter perturbation to the DN decision boundary that forces the model to learn boundaries with increased margins to the closest training samples. From the results that we derived one can directly see how to further improve batch-normalization. For example, by controlling the strength of the noise standard deviation of the batch-normalization parameters to further control the decision boundary margin, or by altering the optimization problem that batch-normalization minimizes to enforce a specific adaptivity of the DN partition to the dataset at hand. We hope that this work will motivate researchers to further extend BN into task-specific methods leveraging a priori knowledge of the properties one desires to enforce into their DN.

## 7 REPRODUCIBILITY STATEMENT

The proofs and further derivations of the theoretical results provided throughout the main text are given in the Appendix. Additional explanatory figures are also provided. The codebase to reproduce not only the quantitative experiments e.g. the performances without learnable $\gamma$ and $\beta$ or the performance of BN used as an initialization only, but also the qualitative ones will be released upon completion of the review process in the hope to increase the theoretical study around BN and to further improve this eponymous technique. We did not consider any sensitive dataset and focused on the standard computer vision ones which are CIFAR variants and Imagenet. For the architectures we also used the standard Resnet50 and Resnet9, additional details are also provided in the Appendix.

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

## A  APPENDIX

We provide in the following appendices details on the Max-Affine Spline formulation of DNs (Sec. B) and the proofs of the various theoretical results from the main text (Sec. E).

## B  DETAILS ON CONTINUOUS PIECEWISE AFFINE DEEP NETWORKS

The goal of this section is to provide additional details into the forming of the per-region affine mappings of CPA DNs.

As mentioned in the main text, any DN that is formed from CPA nonlinearities can be expressed itself as a CPA operator. The per-region affine mappings are thus entirely defined by the *state* of the DN nonlinearities. For an activation function such as ReLU, leaky-ReLU or absolute value, the nonlinearity *state* is completely determined by the sign of the activation input as it determines which of the two linear mapping to apply to produce its output. Let denote this code as $\boldsymbol{q}_\ell(\boldsymbol{z}_{\ell-1}) \in \{\alpha, 1\}^{D_\ell}$ given by

$$[\boldsymbol{q}_\ell(\boldsymbol{z}_{\ell-1})]_i = \begin{cases} \alpha, & [\boldsymbol{W}_\ell \boldsymbol{z}_{\ell-1} + \boldsymbol{b}_\ell]_i \leq 0 \\ 1, & [\boldsymbol{W}_\ell \boldsymbol{z}_{\ell-1} + \boldsymbol{b}_\ell]_i > 0 \end{cases} \quad (14)$$

where the pre-activation formula above can be replaced with the one from (2) if BN is employed. For a max-pooling type of nonlinearity the state corresponds to the argmax obtained in each pooling regions, for details on how to generalize the below in that case we refer the reader to Balestriero & Baraniuk (2018). Based on the above, the layer input-output mapping can be written as

$$\boldsymbol{z}_\ell = \boldsymbol{Q}_\ell(\boldsymbol{z}_{\ell-1})(W_\ell \boldsymbol{z}_{\ell-1} + \boldsymbol{b}_\ell). \quad (15)$$

where $\boldsymbol{Q}_\ell$ produces a diagonal matrix from the vector $\boldsymbol{q}_\ell$, and one has $\alpha = 0$ for ReLU, $\alpha = -1$ for absolute value and $\alpha > 0$ for leaky-ReLU; see Balestriero & Baraniuk (2018) for additional details. The up-to-layer-$\ell$ mapping can thus be easily written as

$$\boldsymbol{z}_\ell = A_{1|\ell}(\boldsymbol{x})\boldsymbol{x} + B_{1|\ell}(\boldsymbol{x}) \quad (16)$$

with the following slope and bias parameters

$$A_{1|\ell}(\boldsymbol{x}) = \boldsymbol{Q}_\ell \boldsymbol{W}_\ell \boldsymbol{Q}_{\ell-1} \boldsymbol{W}_{\ell-1} \dots \boldsymbol{Q}_1 \boldsymbol{W}_1, \quad (17)$$

$$B_{1|\ell}(\boldsymbol{x}) = \sum_{i=1}^{\ell} (\boldsymbol{Q}_\ell \boldsymbol{W}_\ell \boldsymbol{Q}_{\ell-1} \boldsymbol{W}_{\ell-1} \dots \boldsymbol{Q}_{i+1} \boldsymbol{W}_{i+1}) \boldsymbol{b}_i, \quad (18)$$

where for clarity we abbreviated $\boldsymbol{Q}_\ell(\boldsymbol{z}_{\ell-1})$ as $\boldsymbol{Q}_\ell$.

From the above formulation, it is clear that whenever the codes $q_\ell$ stay the same for different inputs, the layer input-output mapping remains linear. This defines a region $\omega_\ell$ of $\Omega_\ell$ from (6), the layer-input-space partition region, as

$$\omega_\ell^{\boldsymbol{q}} = \{\boldsymbol{z}_{\ell-1} \in \mathbb{R}^{D_{\ell-1}} : \boldsymbol{q}_\ell(\boldsymbol{z}_{\ell-1}) = \boldsymbol{q}\}, \boldsymbol{q} \in \{\alpha, 1\}^{D_l}. \tag{19}$$

In a similar way, the up-to-layer-$\ell$ input space partition region can be defined.

The multilayer region is defined as

$$\omega_{1|\ell}^{\boldsymbol{q}_1,\dots,\boldsymbol{q}_\ell} = \bigcap_{i=1}^{\ell} \{\boldsymbol{x} \in \mathbb{R}^D : \boldsymbol{q}_i(\boldsymbol{x}) = \boldsymbol{q}_i\}, \boldsymbol{q}_i \in \{\alpha, 1\}^{D_l}. \tag{20}$$

**Definition 1** (Layer/DN partition)**.** *The layer $\ell$ (resp. -up-to-layer-$\ell$) input space partition is given by*

$$\Omega_\ell = \{\omega_\ell^{\boldsymbol{q}}, \boldsymbol{q} \in \{\alpha, 1\}^{D_l}\} \setminus \emptyset, \tag{21}$$

$$\Omega_{1|\ell} = \{\omega_{1|\ell}^{\boldsymbol{q}_1,\dots,\boldsymbol{q}_\ell}, \boldsymbol{q}_i \in \{\alpha, 1\}^{D_i}, \forall i\} \setminus \emptyset. \tag{22}$$

Note that $\Omega_{1|L}$ forms the entire DN input space partition. Both unit and layer input space partitioning can be rewritten as Power Diagrams, a generalization of Voronoi Diagrams Balestriero et al. (2019). Composing layers then simply refines successively the previously build input space partitioning via a subdivision process to obtain the -up to layer $\ell$- input space partitioning $\Omega_{|\ell}$.

## C  BATCH NORMALIZATION PARAMETER $\boldsymbol{\sigma}$ FOLDS THE PARTITION REGIONS TOWARDS THE TRAINING DATA

We are now in a position to describe the effect of the BN parameter $\boldsymbol{\sigma}_\ell$, which has no effect on the spline partition of an individual DN layer but comes into play for a composition of two or more layers. In contrast to the hyperplane translation effected by $\boldsymbol{\mu}_\ell$, $\boldsymbol{\sigma}_\ell$ optimizes the *dihedral angles* between adjacent facets of the folded hyperplanes in subsequent layers in order to swing them closer to the training data.

Define $\boldsymbol{Q}_\ell$ as the square diagonal matrix whose diagonal entry $i$ is determined by the sign of the pre-activation $h_{\ell,i}$. That is,

$$[\boldsymbol{Q}_\ell]_{i,i} = \begin{cases} \alpha/\sigma_{\ell,i}, & h_{\ell,i} < 0 \\ 1/\sigma_{\ell,i}, & h_{\ell,i} \geq 0, \end{cases} \tag{23}$$

with $\alpha = 0$ for ReLU, $\alpha > 0$ for leaky-ReLU, and $\alpha = -1$ for absolute value (recall the definition of the activation function from Section 2.2; see Balestriero & Baraniuk (2019) for additional non-linearities). Note that $\boldsymbol{Q}_\ell$ is constant across each region $\omega$ in the layer's spline partition $\Omega_\ell$ since, by definition, none of the pre-activations $h_{\ell,i}$ change sign within region $\omega$. We will use $\boldsymbol{Q}_\ell(\omega)$ to denote the dependency on the region $\omega$; to compute $\boldsymbol{Q}_\ell(\omega)$ given $\omega$, one merely samples a layer input from $\omega$, computes the pre-activations $h_{\ell,i}$, and applies (23).

In Appendix **??** we prove that the BN parameter $\boldsymbol{\sigma}_\ell$ adjusts the dihedral folding angles of adjacent facets of each folded hyperplane created by layer $\ell + 1$ in order to align the facets with the training data. Figure 2(c) illustrates empirically how $\boldsymbol{\sigma}_1$ folds the facets of $\mathcal{F}_{2,k}$ realized by the second layer of a toy DN. Since the relevant expressions quickly (but predictably) grow in length with the number of layers, to expose the salient points, but without loss of generality, we will focus the next theorem on the composition of the first two DN layers (layers $\ell = 1, 2$). In this case, there are two geometric quantities of interest that combine to create the input space spline partition $\Omega = \Omega_{|2}$: layer 1's hyperplanes $\mathcal{H}_{1,i}$ and layer 2's folded hyperplanes $\mathcal{F}_{2,k}$.

**Theorem 3.** *Given a 2-layer ($\ell = 1, 2$) BN-equipped DN employing a leaky-ReLU or absolute value activation function, consider two adjacent regions $\omega, \omega'$ from the spline partition $\Omega$ whose boundaries contain the facets $\mathcal{F}_{2,k,\omega}, \mathcal{F}_{2,k,\omega'}$ created by folding across the boundaries' shared hyperplane*

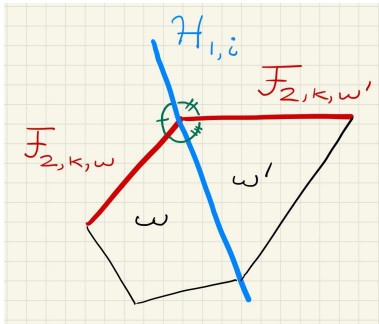

Figure 8: Sketch of the situation in Theorem 3 for a two-dimensional input space.

$\mathcal{H}_{1,i}$ *(see Figure 8). The dihedral angles between these three (partial) hyperplanes are given by:*

$$\theta\left(\mathcal{F}_{2,k,\omega}, \mathcal{H}_{1,i}\right) = \arccos\left(\frac{\left|\boldsymbol{w}_{2,k}^\top \boldsymbol{Q}_1(\omega)\boldsymbol{W}_1\boldsymbol{w}_{1,i}\right|}{\left\|\boldsymbol{w}_{2,k}^\top \boldsymbol{Q}_1(\omega)\boldsymbol{W}_1\right\| \, \left\|\boldsymbol{w}_{1,i}\right\|}\right), \tag{24}$$

$$\theta\left(\mathcal{F}_{2,k,\omega'}, \mathcal{H}_{1,i}\right) = \arccos\left(\frac{\left|\boldsymbol{w}_{2,k}^\top \boldsymbol{Q}_1(\omega')\boldsymbol{W}_1\boldsymbol{w}_{1,i}\right|}{\left\|\boldsymbol{w}_{2,k}^\top \boldsymbol{Q}_1(\omega')\boldsymbol{W}_1\right\| \, \left\|\boldsymbol{w}_{1,i}\right\|}\right), \tag{25}$$

$$\theta\left(\mathcal{F}_{2,k,\omega}, \mathcal{F}_{2,k,\omega'}\right) = \arccos\left(\frac{\left|\boldsymbol{w}_{2,k}^\top \boldsymbol{Q}_1(\omega)\boldsymbol{W}_1\boldsymbol{W}_1^\top \boldsymbol{Q}_1(\omega')\boldsymbol{w}_{2,k}\right|}{\left\|\boldsymbol{w}_{2,k}^\top \boldsymbol{Q}_1(\omega)\boldsymbol{W}_1\right\| \, \left\|\boldsymbol{w}_{2,k}^\top \boldsymbol{Q}_1(\omega')\boldsymbol{W}_1\right\|}\right). \tag{26}$$

In (24)–(26), $\boldsymbol{Q}_1(\omega)$ and $\boldsymbol{Q}_1(\omega')$ differ by only one diagonal entry at index $(i,i)$: one matrix takes the value $\frac{1}{\sigma_{1,i}}$ and the other the value $\frac{\alpha}{\sigma_{1,i}}$, as per Eq. 23. Since (8) implies that $\sigma_{\ell,i}^2 \propto \min_{\boldsymbol{\mu}_{\ell,i}} \mathcal{L}(\boldsymbol{\mu}_{\ell,i}, \mathcal{B}_\ell)$, we have the following two insights that we state for $\ell = 1$. (These results extend in a straightforward fashion for more than two layers as well as for more complicated piecewise linear activation functions.)

On the one hand, if $\mathcal{H}_{1,i}$ is well-aligned with the training data $\mathcal{B}_1$, then the TLS error, and hence $\sigma_{1,i}^2$, will be small. For the absolute value activation function, formulae (24) and (25) then tell us that both $\theta\left(\mathcal{F}_{2,k,\omega}, \mathcal{H}_{1,i}\right) \approx 0$ and $\theta\left(\mathcal{F}_{2,k,\omega'}, \mathcal{H}_{1,i}\right) \approx 0$, meaning that $\mathcal{F}_{2,k,\omega}$ and $\mathcal{F}_{2,k,\omega'}$ will be folded to closely align with $\mathcal{H}_{1,i}$ (and hence the data). The connection between $\sigma_{\ell,i}^2$ and (24,25) lies in the entries of the $\boldsymbol{Q}$ matrix. Basically, the $\boldsymbol{\sigma}_\ell$ are used in the denominator of $\boldsymbol{Q}$ and thus bend more or less the angles. For the ReLU/leaky-ReLU activation function, either $\mathcal{F}_{2,k,\omega}$ or $\mathcal{F}_{2,k,\omega'}$ will be folded to closely align with $\mathcal{H}_{1,i}$; the other facet will be unchanged/mildly folded.

On the other hand, if $\mathcal{H}_{1,i}$ is not well-aligned with the training data $\mathcal{B}_1$, then the TLS error, and hence $\sigma_{1,i}^2$, will be large. This will force $\boldsymbol{Q}_1(\omega) \approx \boldsymbol{Q}_1(\omega')$ and thus $\theta\left(\mathcal{F}_{2,k,\omega}, \mathcal{F}_{2,k,\omega'}\right) \approx \pi$, meaning that a poorly aligned layer-1 hyperplane $H_{1,i}$ will not appreciably fold intersecting layer-2 facets.

Figure 9 illustrates empirically how the BN parameter $\sigma_{\ell,k}$ measures the quality of the fit of the (folded) hyperplanes to the data in the TLS error sense for a toy DN.

In summary, and extrapolating to the general case, the effect of the BN parameter $\boldsymbol{\sigma}_\ell$ is to fold the layer-$(\ell+1)$ hyperplanes (also the $\ell+2$ and subsequent hyperplanes) that contribute to the spline partition boundary $\Omega$ in order to align them with the layer-$\ell$ hyperplanes that match the data well. Hence, not only $\boldsymbol{\mu}_\ell$ but also $\boldsymbol{\sigma}_\ell$ plays a crucial role in aligning the DN spline partition with the training data (recall Figure 1).

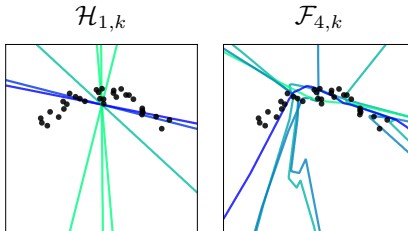

$\mathcal{H}_{1,k}$ $\mathcal{F}_{4,k}$

Figure 9: Layer-1 hyperplanes $\mathcal{H}_{1,k}$ (left) and layer-4 folded hyperplanes $\mathcal{F}_{4,k}$ (right) depicted in the 2D input space of a toy 4-layer DN trained on the data points denoted with black dots. The (folded) hyperplanes are colored based on the corresponding value $\sigma_{\ell,k}^2/\|\boldsymbol{w}_{\ell,k}\|^2$, which is proportional to the total least squares (TLS) fitting error to the data (blue: small error, close to the data points; green: large error, far from the data points).

## D  ROLE OF THE BN PARAMETERS $\boldsymbol{\beta}, \boldsymbol{\gamma}$ AND PROOF OF PROPOSITION 1

Without loss of generality, consider a simple two-layer DN to illustrate. Then, it is clear that $\boldsymbol{\gamma}_1$ simply rescales the rows of $\boldsymbol{W}_2$ and the entries of $\boldsymbol{\beta}_1$

$$\boldsymbol{z}_{2,r} = a\left(\frac{\sum_{k=1}^{D_1} [\boldsymbol{w}_{2,r}]_k\, a\left(\frac{\langle \boldsymbol{w}_{1,k}, \boldsymbol{x}\rangle - \mu_{1,k}}{\sigma_{1,k}}\gamma_{1,k} + \beta_{1,k}\right) - \mu_{2,r}}{\sigma_{2,r}}\gamma_{2,r} + \beta_{2,r}\right) \tag{27}$$

$$= a\left(\frac{\sum_{k=1}^{D_1} \gamma_{1,k}[\boldsymbol{w}_{2,r}]_k\, a\left(\frac{\langle \boldsymbol{w}_{1,k}, \boldsymbol{x}\rangle - \mu_{1,k}}{\sigma_{1,k}} + \frac{\beta_{1,k}}{\gamma_{1,k}}\right) - \mu_{2,r}}{\sigma_{2,r}}\gamma_{2,r} + \beta_{2,r}\right) \tag{28}$$

$$= a\left(\frac{\sum_{k=1}^{D_1} [\boldsymbol{w}'_{2,r}]_k\, a\left(\frac{\langle \boldsymbol{w}_{1,k}, \boldsymbol{x}\rangle - \mu_{1,k}}{\sigma_{1,k}} + \beta'_{1,k}\right) - \mu_{2,r}}{\sigma_{2,r}}\gamma_{2,r} + \beta_{2,r}\right) \tag{29}$$

and so does not need to be optimized separately. Here, $[\boldsymbol{w}_{2,r}]_k$ denotes the $k$-th entry of the $r$-th row of $\boldsymbol{W}_2$. We obtain (28), because standard activation and pooling functions (e.g., (leaky-)ReLU, absolute value, max-pooling) are such that $a(cu) = c\,a(u)$. This leaves $\boldsymbol{\beta}_\ell$ as the only learnable parameter that needs to be considered.

Now, our BN theoretical analysis relies on setting $\boldsymbol{\beta}_\ell = \boldsymbol{0}$, which corresponds to the standard initialization of BN. In this setting, we saw that BN "fits" the partition boundaries to the data samples exactly. Now, by observing the form of (2), it is clear that learning $\boldsymbol{\beta}_\ell$ enables to "undo" this fitting if needed to better solve the task-at-hand. However, we have found that in most practical scenarios, fixing $\boldsymbol{\beta}_\ell = \boldsymbol{0}$ throughout training actually does not alter final performances. In fact, on Imagenet, the top1 accuracy of a Resnet18 goes from 67.60% to 65.93% when doing such a change, and on a Resnet34, from 68.71% to 67.21%. The drop seems to remain the same even on more complicated architecture as for a Resnet50, the top1 test accuracy goes down from 77.11% to 74.98%, where again we emphasize that the exact same hyper-parameters are employed for both situations i.e. the drop could potentially be reduced by tuning the hyper-parameters. Obviously those numbers might vary depending on optimizers and other hyper-parameters, we used here the standard ones for those architectures since our point is merely that all our theoretical results relying on $\boldsymbol{\beta}_\ell = \boldsymbol{0}, \boldsymbol{\gamma}_\ell = \boldsymbol{1}$ still applies to high performing models.

In addition to the above Imagenet results, we provide in Table 1 the results for the classification accuracy on various dataset and with two architectures. This set of results simply demonstrates that the learnable parameters of BN ($\gamma, \beta$) have very little impact of performances.

Table 1: Test accuracy of various models when employing (yes) or not (no) the BN learnable parameters, as was demonstrated in the main paper, those parameters have very little impact on the final test accuracy (no data-augmentation is used).

|  | imagenette | | cifar10 | | cifar100 | | svhn | |
|---|---|---|---|---|---|---|---|---|
|  | No | Yes | No | Yes | No | Yes | No | Yes |
| RESNET | 79.2 | 78.8 | 83. | 86.2 | 50. | 54.8 | 94.2 | 95.3 |
| CNN | 77.7 | 77.6 | 87.5 | 87.6 | 54. | 55.2 | 96. | 95.9 |

We provide below the descriptions of the DN architectures. For the Residual Networks, the notation Resnet4-10 for example defines the width factor and depth of each block. We provide an example below for Resnet2-2.

### Residual Network

```
Conv2D(layer[-1], 32, (5, 5), pad="SAME", b=None))
BatchNorm(layer[-1])
leaky_relu(layer[-1])
MaxPool2D(layer[-1], (2, 2))
Dropout(layer[-1], 0.9)

for width in [64, 128, 256, 512]:
    for repeat in range(2):
        Conv2D(layer[-1], width, (3, 3), b=None, pad="SAME")
        BatchNorm(layer[-1])
        leaky_relu(layer[-1])
        Conv2D(layer[-1], width, (3, 3), b=None, pad="SAME")
        BatchNorm(layer)
        if layer[-6].shape == layer[-1].shape:
            leaky_relu(layer[-1]) + layer[-6])
        else:
            leaky_relu(layer[-1])
                + Conv2D(layer[-6], width, (3, 3),
                b=None, pad="SAME")

    AvgPool2D(layer[-1], (2, 2))
    Conv2D(layer[-1], 512, (1, 1), b=None))
    BatchNorm(layer)
    leaky_relu(layer[-1])

GlobalAvgPool2D(layer[-1])
Dense(layer[-1], N_CLASSES)
```

We now describe the CNN model that we employed. Notice that if the considered dataset is imagenette or other (smaller spatial dimension) dataset there is an additional first layer of convolution plus spatial pooling to reduce the spatial dimensions of the feature maps.

### Convolutional Network

```
Conv2D(layer[-1], 32, (5, 5), pad="SAME", b=None)
BatchNorm(layer)
leaky_relu(layer[-1]))
MaxPool2D(layer[-1], (2, 2))

if args.dataset == "imagenette":
    Conv2D(layer[-1], 64, (5, 5), pad="SAME", b=None)
    BatchNorm(layer)
    leaky_relu(layer[-1])
    MaxPool2D(layer[-1], (2, 2))

for k in range(3):
    Conv2D(layer[-1], 96, (5, 5), b=None, pad="SAME")
    BatchNorm(layer)
    leaky_relu(layer[-1])
    Conv2D(layer[-1], 96, (1, 1), b=None)
    BatchNorm(layer)
    leaky_relu(layer[-1])
    Conv2D(layer[-1], 96, (1, 1), b=None)
    BatchNorm(layer)
```

```
    leaky_relu(layer[-1])

Dropout(layer[-1], 0.7)
MaxPool2D(layer[-1], (2, 2))

for k in range(3):
    Conv2D(layer[-1], 192, (5, 5), b=None, pad="SAME")
    BatchNorm(layer)
    leaky_relu(layer[-1])
    Conv2D(layer[-1], 192, (1, 1), b=None)
    BatchNorm(layer)
    leaky_relu(layer[-1])
    Conv2D(layer[-1], 192, (1, 1), b=None)
    BatchNorm(layer)
    leaky_relu(layer[-1])

Dropout(layer[-1], 0.7)
MaxPool2D(layer[-1], (2, 2))

Conv2D(layer[-1], 192, (3, 3), b=None)
BatchNorm(layer)
leaky_relu(layer[-1])
Conv2D(layer[-1], 192, (1, 1), b=None))
BatchNorm(layer)
leaky_relu(layer[-1])

GlobalAvgPool2D(layer[-1])
Dense(layer[-1], N_CLASSES)
```

# E  PROOFS

We propose in this section the various proofs supporting the diverse theoretical claims from the main part of the paper.

## E.1  PROOF OF THEOREM 1

*Proof.* In order to prove the theorem we will demonstrate below that the optimum of the total least square optimization problem is reached at the unique global optimum given by the average of the data, hence corresponding to the batch-normalization mean parameter. Then we demonstrate that at this minimum, the value of the total least square loss is given by the variance parameter of batch-normalization.

The optimization problem is given by

$$\mathcal{L}(\mu; Z) = \sum_{k=1}^{D_\ell} \sum_{\boldsymbol{z} \in Z} d\left(\boldsymbol{z}, \mathcal{H}_{\ell,k}\right)^2 = \sum_{k=1}^{D_\ell} \sum_{\boldsymbol{z} \in Z} \frac{\left|\langle \boldsymbol{w}_{\ell,k}, \boldsymbol{z}_{\ell-1} \rangle - \boldsymbol{\mu}_k\right|^2}{\|\boldsymbol{w}_{\ell,k}\|_2^2} \tag{30}$$

it is clear that the optimization problem

$$\min_{\mu \in \mathbb{R}^{D_\ell}} \mathcal{L}(\mu, Z), \tag{31}$$

can be decomposed into multiple independent optimization problem for each dimension of the vector $\mu$, since we are working with an unconstrained optimization problem with separable sum. We thus focus on a single $\boldsymbol{\mu}_k$ for now. The optimization problem becomes

$$\min_{\boldsymbol{\mu}_k \in \mathbb{R}} \sum_{\boldsymbol{z} \in Z} \frac{\left|\langle \boldsymbol{w}_{\ell,k}, \boldsymbol{z}_{\ell-1} \rangle - \boldsymbol{\mu}_k\right|^2}{\|\boldsymbol{w}_{\ell,k}\|_2^2} \tag{32}$$

taking the first derivative leads to

$$\partial \sum_{\boldsymbol{z} \in Z} \frac{|\langle \boldsymbol{w}_{\ell,k}, \boldsymbol{z}_{\ell-1} \rangle - \boldsymbol{\mu}_k|^2}{\|\boldsymbol{w}_{\ell,k}\|_2^2} = -2 \sum_{\boldsymbol{z} \in Z} \frac{(\langle \boldsymbol{w}_{\ell,k}, \boldsymbol{z}_{\ell-1} \rangle - \boldsymbol{\mu}_k)}{\|\boldsymbol{w}_{\ell,k}\|_2^2} \tag{33}$$

$$= -2 \sum_{\boldsymbol{z} \in Z} \frac{\langle \boldsymbol{w}_{\ell,k}, \boldsymbol{z}_{\ell-1} \rangle}{\|\boldsymbol{w}_{\ell,k}\|_2^2} + 2 Card(Z) \frac{\boldsymbol{\mu}_k}{\|\boldsymbol{w}_{\ell,k}\|_2^2} \tag{34}$$

$$\tag{35}$$

the above first derivative of the total least square (quadratic) loss function is thus a linear function of $\boldsymbol{\mu}_k$ being 0 at the unique point given by

$$-2 \sum_{\boldsymbol{z} \in Z} \frac{\langle \boldsymbol{w}_{\ell,k}, \boldsymbol{z}_{\ell-1} \rangle}{\|\boldsymbol{w}_{\ell,k}\|_2^2} + 2 Card(Z) \frac{\boldsymbol{\mu}_k}{\|\boldsymbol{w}_{\ell,k}\|_2^2} = 0 \iff \boldsymbol{\mu}_k = \frac{\sum_{\boldsymbol{z} \in Z} \langle \boldsymbol{w}_{\ell,k}, \boldsymbol{z}_{\ell-1} \rangle}{Card(Z)} \tag{36}$$

confirming that the average of the pre-activation feature maps (per-dimension) is indeed the optimum of the optimization problem. One can verify easily that it is indeed a minimum by taking the second derivative of the total least square which indeed positive and given by $\frac{2 Card(Z)}{\|\boldsymbol{w}_{\ell,k}\|_2^2}$. The above can be done for each dimension $k$ in a similar manner. Now, by inserting this optimal value back into the total least square loss, we obtain the desired result. □

### E.2 PROOF OF CENTRAL HYPERPLANE ARRANGEMENT

**Corollary 2.** *BN constrains the input space partition boundaries $\partial \Omega_\ell$ of each layer $\ell$ of a BN-equipped DN to be a central hyperplane arrangement; indeed, the average of the layer's training data inputs*

$$\overline{\boldsymbol{z}}_\ell \in \bigcap_{k=1}^{D_\ell} \mathcal{H}_{\ell,k} \tag{37}$$

*as long as $\|\boldsymbol{w}_{\ell,k}\| > 0$.*

*Proof.* In order to prove the desired result i.e. that there exists a nonempty intersection between all the hyperplanes, we first demonstrate that the layer input centroid $\overline{\boldsymbol{z}}_{\ell-1}$ indeed to one hyperplane, say $k$. Then it will be direct to see that this holds regardless of $k$ and thus the intersection of all hyperplanes contains at least $\overline{\boldsymbol{z}}_{\ell-1}$ which is enough to prove the statement.

For a data point (in our case $\overline{\boldsymbol{z}}_{\ell-1}$) to belong to the $k^{\text{th}}$ (unit) hyperplane $\mathcal{H}_{\ell,k}$ of layer $\ell$, we must ensure that this point belong to the set of the hyperplane defined as (recall (5))

$$\mathcal{H}_{\ell,k} = \left\{ \boldsymbol{z}_{\ell-1} \in \mathbb{R}^{D_{\ell-1}} : \langle \boldsymbol{w}_{\ell,k}, \boldsymbol{z}_{\ell-1} \rangle = [\mu_\ell]_k \right\}, \tag{38}$$

in our case we can simply use the data centroid and ensure that it fulfils the hyperplane equality

$$\langle \boldsymbol{w}_{\ell,k}, \overline{\boldsymbol{z}}_{\ell-1} \rangle = \left\langle \boldsymbol{w}_{\ell,k}, \frac{\sum_{\boldsymbol{z} \in Z} \boldsymbol{z}}{Card(Z)} \right\rangle = \sum_{\boldsymbol{z} \in Z} \frac{\langle \boldsymbol{w}_{\ell,k}, \boldsymbol{z} \rangle}{Card(Z)} = [\mu_\ell^*]_k \tag{39}$$

where the last equation gives in fact the batch-normalization mean parameter. So now, recalling the equation of $\mathcal{H}_{\ell,k}$ we see that the point $\overline{\boldsymbol{z}}_{\ell-1}$ makes plane projection $[\mu_\ell^*]_k$ which equals the bias of the hyperplane effectively making $\overline{\boldsymbol{z}}_{\ell-1}$ part of the (batch-normalized) hyperplane $\mathcal{H}_{\ell,k}$. Doing the above for each $k \in D_\ell$ we see that the layer input centroid belongs to all the unit hyperplane that are shifted by the correct batch-normalization parameter, hence we directly obtain the desired result

$$\overline{\boldsymbol{z}}_{\ell-1} \subset \bigcap_{k \in D_\ell} \mathcal{H}_{\ell,k}, \tag{40}$$

concluding the proof. □

### E.3 PROOF OF THEOREM 2

*Proof.* Define by $\boldsymbol{x}^*$ the shortest point in $\mathcal{P}_{\ell,k}$ from $\boldsymbol{x}$ defined by

$$\boldsymbol{x}^* = \arg\min_{\boldsymbol{u} \in \mathcal{P}_{\ell,k}} \|\boldsymbol{x} - \boldsymbol{u}\|_2.$$

The path from $\boldsymbol{x}$ to $\boldsymbol{x}^*$ is a straight line in the input space which we define by

$$l(\theta) = \boldsymbol{x}^*\theta + (1-\theta)\boldsymbol{x}, \theta \in [0,1], \tag{41}$$

s.t. $l(0) = \boldsymbol{x}$, our original point, and $l(1)$ is the shortest point on the kinked hyperplane. Now, in the input space of layer $\ell$, this parametric line becomes a continuous piecewise affine parametric line defined as

$$\boldsymbol{z}_{\ell-1}(\theta) = (f_{\ell-1} \circ \cdots \circ f_1)(l(\theta)). \tag{42}$$

By definition, if $\mathcal{P}_{\ell,k}$ is brought closer to $\boldsymbol{x}$, it means that $\exists\theta < 1$ s.t. $l(\theta) \in \mathcal{P}_{\ell,k}$. Similarly this can be defined in the layer input space as follows.

$$\exists\theta' < 1 \ \ s.t. \ \ \boldsymbol{z}_{\ell-1}(\theta) \in \mathcal{H}_{\ell,k} \implies \exists\theta < 1 \ \ s.t. \ \ l(\theta) \in \mathcal{P}_{\ell,k}$$

this demonstrates that when moving the layer hyperplane s.t. it intersects the kinked path $\boldsymbol{z}_{\ell-1}$ at a point $\boldsymbol{z}_{\ell-1}(\theta')$ with $\theta' < 1$, then the distance in the input space is also reduced. Now, the BN fitting is greedy and tried to minimize the length of the straight line between $\boldsymbol{z}_{\ell-1}(0)$ a.k.a $\boldsymbol{z}_{\ell-1}(\boldsymbol{x})$ and the hyperplane $\mathcal{H}_{\ell,k}$. However, notice that if the length of this straight line decreases by brining the hyperplane closer to $\boldsymbol{z}_{\ell-1}(\boldsymbol{x})$ then this also decreases the $\theta'$ s.t. $\boldsymbol{z}_{\ell-1}(\theta') \in \mathcal{H}_{\ll,k}$ in turn reducing the distance between $\boldsymbol{x}$ and $\mathcal{P}_{\ell,k}$ in the DN input space, giving the desired (second) result. Conversely, if $\boldsymbol{z}_{\ell-1}(0) \in \mathcal{H}_{\ell,k}$ then the point $\boldsymbol{x}$ lies in the zero-set of the unit, in turn making it belong to the kinked hyperplane $\mathcal{P}_{\ell,k}$ which corresponds to this exact set.

□

### E.4 PROOF OF PROPOSITION 3

**Proposition 3.** *Consider an L-layer DN configured to learn a binary classifier from the labeled training data $\mathcal{X}$ using a leaky-ReLU activation function, arbitrary weights $\boldsymbol{W}_\ell$ at all layers, BN at layers $1, \ldots, L-1$, and layer $L$ configured as in (1) with $\boldsymbol{c}_L = \boldsymbol{0}$. Then, for any training mini-batch from $\mathcal{X}$, there will be at least one data point on either side of the decision boundary.*

*Proof.* When using leaky-ReLU the input to the last layer will have positive and negative values in each dimension for at least 1 in the current minibatch. That means that each dimension will have at least 1 negative value and all the other positive or vice-versa. As the last layer is initialized with zero bias, the decision boundary is defined in the last layer input space as the hyperplanes (or zero-set) of each output unit. Also, being on one side or the other of the decision boundary in the DN input space is equivalent to being on one side or the other of the linear decision boundary in the last layer input space. Combining those two results we obtain that at initialization, there has to be at least 1 sample one side of the decision boundary and the others on the other side. □

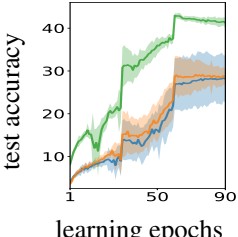

Figure 10: Image classification using a Resnet9 on CIFAR100. No BN or data augmentation was used during SGD training. Instead, the DN was initialized with random weights and zero bias (blue), random bias (orange), or via BN across the entire data set as in Figure 3 (green). Each training was repeated 10 times with learning rate cross-validation, and we plot the average test accuracy (of the best valid set learning rate) vs. learning epoch. BN's "smart initialization" reaches a higher-performing solution faster because it provides SGD with a spline partition that is already adapted to the training dataset.

### E.5 Proof of BN Statistics Variance

*Proof.* Let's consider a random variable with $E(\boldsymbol{z}) = \boldsymbol{m}$ and $Cov(\boldsymbol{z}) = \mathrm{diag}(\rho^2)$. Then we directly have that

$$
\begin{aligned}
E(\langle \boldsymbol{w}, \boldsymbol{z} \rangle) =& \langle \boldsymbol{w}, \boldsymbol{m} \rangle \\
Var(\langle \boldsymbol{w}, \boldsymbol{z} \rangle) =& E((\langle \boldsymbol{w}, \boldsymbol{z} \rangle - E(\langle \boldsymbol{w}, \boldsymbol{z} \rangle))^2) \\
=& E((\langle \boldsymbol{w}, \boldsymbol{z} \rangle - \langle \boldsymbol{w}, \boldsymbol{m} \rangle)^2) \\
=& E(\langle \boldsymbol{w}, \boldsymbol{z} - \boldsymbol{m} \rangle^2) \\
=& E\left( \sum_d \boldsymbol{w}_d^2 (\boldsymbol{z}_d - \boldsymbol{m}_d)^2 + \sum_{d \neq d'} \boldsymbol{w}_d (\boldsymbol{z}_d - \boldsymbol{m}_d) \boldsymbol{w}_{d'} (\boldsymbol{z}_{d'} - \boldsymbol{m}_{d'}) \right) \\
=& E\left( \sum_d \boldsymbol{w}_d^2 (\boldsymbol{z}_d - \boldsymbol{m}_d)^2 + \sum_{d \neq d'} \boldsymbol{w}_d (\boldsymbol{z}_d - \boldsymbol{m}_d) \boldsymbol{w}_{d'} (\boldsymbol{z}_{d'} - \boldsymbol{m}_{d'}) \right) \\
=& \sum_d \boldsymbol{w}_d^2 \rho_d^2 \\
=& \langle \boldsymbol{w}^2, \rho_d^2 \rangle
\end{aligned}
$$

now given the known variance and mean of the $\langle \boldsymbol{w}, \boldsymbol{z} \rangle$ random variable, the desired result is obtained by using the standard empirical variance estimator. $\qquad \square$

## F Dataset Descriptions

**MNIST** The MNIST database (Modified National Institute of Standards and Technology database) is a large database of handwritten digits that is commonly used for training various image processing systems. The database is also widely used for training and testing in the field of machine learning. It was created by "re-mixing" the samples from NIST's original datasets. The creators felt that since NIST's training dataset was taken from American Census Bureau employees, while the testing dataset was taken from American high school students, it was not well-suited for machine learning experiments. Furthermore, the black and white images from NIST were normalized to fit into a $28x28$ pixel bounding box and anti-aliased, which introduced grayscale levels.

The MNIST database contains $60,000$ training images and $10,000$ testing images. Half of the training set and half of the test set were taken from NIST's training dataset, while the other half of the training set and the other half of the test set were taken from NIST's testing dataset. The original creators of the database keep a list of some of the methods tested on it. In their original paper, they use a support-vector machine to get an error rate of $0.8\%$.

**SVHN** SVHN is a real-world image dataset for developing machine learning and object recognition algorithms with minimal requirement on data preprocessing and formatting. It can be seen as similar in flavor to MNIST (e.g., the images are of small cropped digits), but incorporates an order of magnitude more labeled data (over $600,000$ digit images) and comes from a significantly harder, unsolved, real world problem (recognizing digits and numbers in natural scene images). SVHN is obtained from house numbers in Google Street View images.

**CIFAR10**The CIFAR-10 dataset (Canadian Institute For Advanced Research) is a collection of images that are commonly used to train machine learning and computer vision algorithms. It is one of the most widely used datasets for machine learning research. The CIFAR-10 dataset contains $60,000$ $32x32$ color images in 10 different classes. The 10 different classes represent airplanes, cars, birds, cats, deer, dogs, frogs, horses, ships, and trucks. There are 6,000 images of each class.

Computer algorithms for recognizing objects in photos often learn by example. CIFAR-10 is a set of images that can be used to teach a computer how to recognize objects. Since the images in CIFAR-10 are low-resolution $(32x32)$, this dataset can allow researchers to quickly try different algorithms to see what works. Various kinds of convolutional neural networks tend to be the best at recognizing the images in CIFAR-10.

**CIFAR100**This dataset is just like the CIFAR-10, except it has 100 classes containing 600 images each. There are 500 training images and 100 testing images per class. The 100 classes in the CIFAR-100 are grouped into 20 superclasses. Each image comes with a "fine" label (the class to which it belongs) and a "coarse" label (the superclass to which it belongs).

