# OpenReview forum: "Batch Normalization Explained"
_ICLR.cc/2023/Conference — Submitted to ICLR 2023_

### Official Review · Reviewer_3myR · 2022-10-21

**Confidence:** 4
**Correctness:** 3
**Technical Novelty And Significance:** 2
**Empirical Novelty And Significance:** Not applicable
**Recommendation:** 3

**Clarity, Quality, Novelty And Reproducibility:**

### Quality

 - Theorem&nbsp;1 states that the distance between a set of data points and the boundaries of the splines (i.e. the network layer) is minimised by shifting each pre-activation by its negative mean.
   I am wondering whether this theorem could not be formulated without splines, e.g. in terms of decision boundaries.
   In other words, it is not entirely clear to me what the added value is of using the spline formulation.
 - I was unable to make sense of the proof for Theorem&nbsp;2.
   Unless this proof actually uses results from spline theory, I would expect that this result could also be obtained outside of the context of splines.
 - In corollary&nbsp;1, only the $\Rightarrow$-part would be a corollary of theorem&nbsp;2.
   I do not see how the $\Leftarrow$-part would follow from theorem&nbsp;2.
 - Most of the experiments focus on the effect of the bias parameter.
   I believe it would be useful to also include a setting where results are compared with learned biases.
   After all, a network might be able to learn that the decision boundary should be close to the mean.
   As a matter of fact, it is not uncommon to initialise the biases with the mean of the signal (e.g. Mishkin & Matas, 2015; Karpathy, 2019).
 - The final paragraph of section&nbsp;5 should include a significance test of some sort.
   The improvements look very much like they could be due to randomness.

### Clarity
 - I have a hard time understanding what Figure&nbsp;4.
   First of all, I have no idea what the values on the $y$-axis really mean.
   Instead of using raw counts, a density approach (similar to Figure&nbsp;3) would probably be more informative.
   Furthermore, I do not see why *BN only focused the partition around the CIFAR images and not around the random ones*.
   To me, it seems like there is generally less focusing around random images than around CIFAR images.
   On the other hand, BN consistently focuses more than other approaches.
   Maybe I am missing something, but I do not see what BN is doing special and/or better, here.
 - Should $\beta_\ell$ be $\sigma_\ell$ in the penultimate paragraph of section&nbsp;4?
   If not, what happens to the standard deviation when using BN as initialisation?
 - I do not understand how the smart initialisation is related to the spline analysis.
   Moreover, insights about this effect seem to have no real practical consequences for BN.
   It merely advocates for using LSUV initialisation (Mishkin & Matas, 2015).
 - Similarly, I do not see a connection between Proposition&nbsp;2 and the spline analysis of BN.
   The experiment by itself looks interesting and I can not think of a paper that actually tested this.
   However, I do not understand what this has to do with splines.

### Originality
 - It could be argued that centring the data obviously reduces the distance to decision boundaries going through the origin.
   In this sense, it could be argued that this paper does not really provide new insights since that is exactly what BN was designed to do.
 - Is using BN as initialisation not exactly the same as LSUV initialisation (Mishkin & Matas, 2015)?

### Minor Comments

 - Leaky-RELU, which should be $\max(\alpha x, x)$ with $\alpha \in (0, 1]$, is typically attributed to (Maas et al., 2013)
 - Absolute value is not really a typical example of an activation function.
 - Instead of distinguishing between different activation function, it might be cleaner to simply consider a general function of the form $\phi(x) = \begin{cases} x & x \geq 0 \\ \alpha x & x < 0 \end{cases}$.
 - I am not sure if I understand why eq.&nbsp;(9) is the objective of a total least square problem.
   Also, I would argue that it is not really relevant for the point the authors are trying to make.
   Maybe it would be less confusing to move this to a footnote, to avoid confusion.
 - Typo after footnote&nbsp;3.
 - Theorem&nbsp;2 states that weight matrices should be fixed twice.
 - Typo in corollary&nbsp;1: if and only **if**.
 - Figure&nbsp;3 seems to list facets for layers $j = 3, 7, 11$ (rather than, $j = 1, 7, 11$ as indicated in the main text and caption).
   Alternatively, the image labels are wrong.
 - Color-blind people will probably have trouble understanding figures&nbsp;6 and&nbsp;7.
 - The proofs are generally quite sloppy (different notation, missing factors, ...).

### References

 - Maas, A. L., Hannun, A. Y., & Ng, A. Y. (2013, June). Rectifier Nonlinearities Improve Neural Network Acoustic Models. Workshop for Deep Learning for Audio, Speech, and Language Processing, ICML, Atlanta, GA, USA.
 - Karpathy A. (2019, April). A Recipe for Training Neural Networks. [Blogpost.](http://karpathy.github.io/2019/04/25/recipe/#2-set-up-the-end-to-end-trainingevaluation-skeleton--get-dumb-baselines)

**Strength And Weaknesses:**

- [+] The paper is generally well-written and easy to follow.
- [+] The spline formulation to study BN is a refreshing idea.
- [+] Most of the theoretical derivations seem to be correct.
- [+] The experiment in section&nbsp;5 nicely illustrates how batch noise maximises the margin around the decision boundary.
- [-] I do not see the connection between the spline interpretation and the benefit sections (section&nbsp;4 and&nbsp;5).
- [-] There seem to be no real *new* insights from the spline interpretation.
- [-] This paper seems to be more about bias parameters than about BN.

**Summary Of The Paper:**

This work aims to provide insights into Batch Normalisation (BN).
This is done by formulating Neural Networks as piece-wise affine splines.
It is shown that the mean shift in BN moves the spline boundaries closer to the training data.
Furthermore, two benefits of BN are highlighted.
The first benefit is that the mean shift provides a better initialisation for the bias, compared to standard approaches.
The second benefit is the stochasticity due to the use of mini-batch statistics, which leads to larger margins.

**Summary Of The Review:**

Although this work presents a few interesting ideas, I do not think that this paper provides enough *new* insights to be valuable for the community.
The main takeaway for me would have been that neural networks with LeakyRelus can be interpreted as splines, but this has been established in prior work.
The fact that sections&nbsp;4 and&nbsp;5 appear to be separated from the actual analysis in sections&nbsp;2 and&nbsp;3 seems to be a strong indication that the applicability of the presented analysis is limited.

---

### Official Review · Reviewer_jK7q · 2022-10-23

**Confidence:** 3
**Correctness:** 4
**Technical Novelty And Significance:** 3
**Empirical Novelty And Significance:** 3
**Recommendation:** 6

**Clarity, Quality, Novelty And Reproducibility:**

This constitutes potentially a significant contribution in the community, as BN is widely used and the current paper explains some of the benefits that offers. The writing is clear and I think that only few things can be improved (see questions).

**Strength And Weaknesses:**

I like the paper because it places a simple question and describes the answer in a rather accessible way. I think that the motivation is clear and the idea easy to grasp. The claims are supported by theoretical results, which I have not checked in details, and empirical demonstrations.

**Summary Of The Paper:**

The authors study the effect of batch normalization on the properties of the associated function. In particular, they consider piecewise affine deep neural networks and show that applying BN brings the induced convex polytopes in the input space closer to the training data, which enhances faster training. Consequently, BN can be seen as a practical way to initialize the model. Also, a second observation is that the batches induce some kind of stochasticity in the model, since the resulting function is different for each batch. This potentially helps the model to generalize better. The arguments are supported by theoretical argumentation and empirical demonstrations.

**Summary Of The Review:**

Questions:

1. In Sec 2.3 it is shown that in a feature space (the output of a hidden layer) each row of the next weight matrix introduces a hyperplane. The argument is that BN brings this hyperplane closer to the points in this feature space. I guess the implication is that the associated folded hyperplane in the input space moves closer to the data as well. However, I think that this implication is not explained clearly, since it is only shown that the hyperplane in the feature space moves closer to the points therein, and it is not discussed the relationship to the input space.

2. I am not sure why Theorem 2 is necessary, since the behavior of the folded hyperplane moving closer to the training data is implied by the fact that the hyperplane in the feature space moves closer to the points therein. Is there some detail that I am missing?

3. I think that in Sec 3.1 the explanation of the folded hyperplanes is a bit hard to access. I believe that some figures to illustrate the information and the pipeline could help. I acknowledge that the intuition is clear i.e. a hyperplane in a feature space corresponds to a folded hyperplane in the input space and the collection them for all the layes partitions the input space. Also, there might be some  problems with notation e.g. in Eq. 10 the $D_{j}$ seems weird since for $j=1$ it implies that the weight matrix multiplied with the inputs has only $D_j$ rows. Similarly at the end of this page the $\cup_{\omega\in|\Omega_j}$ perhaps should be $|\Omega_\ell$?

4. As regards Sec 4, if I understood correctly you just make one forward pass (using a batch or all data) for the randomly initialized network, you compute the BN parameters which are kept fixed for the rest of the training?

5. In Sec 5, the layer input $z_\ell$ in the theorem are the features of a single batch or the collection of many different batches?  Also, the results in Fig. 6 imply that a model is trained with BN, and after that a forward pass for every batch gives a different decision boundary?

In general, the style of the paper as it describes in an accessible way some interesting properties  of a technique that is widely used. The analysis is quite easy to follow and the arguments are supported both by theoretical results and empirical demonstrations. Perhaps some minor changes can be mode to improve even further the accessibility of the manuscript.

---

### Official Review · Reviewer_991a · 2022-10-24

**Confidence:** 4
**Correctness:** 1
**Technical Novelty And Significance:** 2
**Empirical Novelty And Significance:** 3
**Recommendation:** 3

**Clarity, Quality, Novelty And Reproducibility:**


Many statements are not clear. For example, adaption, geometry of splines, matching data, margin, increasing margin with batch normalization, smart initialization. Developing a smart initialization technique using batch norm does not conclude batch norm is a smart initialization technique



**Strength And Weaknesses:**

**Strength**

I think that the strongest message of this paper relates to smart initialization. I wished the authors develop an argument around this result.

**Weakness**
- *claims* The paper is full of vague claims that are not supported by theoretical or at least experimental pieces of evidence.
  - It is claimed that BN adapts the geometry of a DN's spline partition. Which theorem or proposition does imply an adaption of spline? I believe that the result of Theorem 1 is a simple fact that the average of points has the minimum square distance to a set of points, which does not relate to adapting partition
  - It is claimed that adapts the geometry partition to **match the data**. Consider a network with random weights. In each layer, partitions are highly random in each layer. How even it is possible to achieve matched data partitioning? What is exactly the meaning of matching, geometry, and adapting here?

 - *soundness* Proposition 1 states that BN parameters can be frozen without losing expressive power. The proof neglects that the variance changes with scaling the weight matrix. Hence, the statement and the proof are not correct.
- *novelty* Thm 1 is only proving the average achieves the minimum squared distance from a set of points. On the other hand, Proposition 2 does not imply an increase in the margin. There is a large gap between what is claimed and what is actually proven.
- *notations* In equation 5, is $z_\ell$ the same as those in equation 3? I believe that equation 5 requires a different notation.
- *literature review* There is recent research on the role of batch normalization for randomly initialized neural nets. The notion of spline adaption may be connected to one of the well-defined properties studied in this literature.
- *experiments* To show BN is equivalent to a smart initialization technique, we need to include BN in Fig 5. Does author use the same batchsize used in BN (often of size 32) to implement the smart initialization?

**Summary Of The Paper:**

This paper targets the inner working of batch normalization. The paper presents two different claims for enhanced learning with batch normalization:
- *unsupervised*  Blindness to the labels batch normalizes adapts the "adapts the geometry of a DN’s spline partition
to match the data" Apparently, this claim relates to theoretical results which justify the mean-deduction in batch normalization is the minimizer of a function.
- *supervised* BN can be viewed as a smart initialization which facilitates classification

**Summary Of The Review:**

I recommend authors to develop  *smart initialization* results and make it either theoretical or experimentally more sound and convincing.

---

### Official Review · Reviewer_maYa · 2022-10-26

**Confidence:** 5
**Correctness:** 2
**Technical Novelty And Significance:** 2
**Empirical Novelty And Significance:** 2
**Recommendation:** 3

**Clarity, Quality, Novelty And Reproducibility:**

There are a few writing issues I would like to point out.
1. Lots of claims are very ambiguous, as mentioned in Weaknesses 2 and 3. It would be good if the authors could define everything more explicitly in math.
2. There is a big logical jump between the key finding and the two benefits. In my understanding, Benefit 1 is a consequence of the key finding, but Benefit 2 is something not related to the key finding.
3. The title suggests that BN can be fully explained, which is inappropriate. As the authors themselves point out in the abstract, this paper only studies BN from the perspective of function approximation. There are also many perspectives, including the works on scale-invariance mentioned in the introduction.

**Strength And Weaknesses:**

#### Strength:
1. It is a good angle to understand BN from the perspective of function approximation.

#### Weakness:
1. The biggest weakness of this paper is that it first fixes gamma and beta parameters in BN to 1 and 0, then claims that the benefits of BN come from the fact that the output of each BN is normalized to mean 0 and std 1, hence concluding that BN makes the weights minimize a TLS distance in some sense. This logic does not make sense because this minimization does not occur when gamma and beta parameters are not fixed. I agree with the authors that gamma and beta can be fixed without hurting performance, but then the situation looks like this: with gamma and beta unfixed, performance is good, and the minimization does not occur; with gamma and beta fixed, performance is nearly the same, and the minimization occurs. Comparing these two cases, a reasonable conclusion is that it is fixing gamma and beta that makes the minimization happens, not BN; and this minimization does not have much effect on performance.
2. Benefit 1 is very ambiguous. In my understanding, the claim that BN is a "smart" initialization means that the initialization of a neural net with BN makes the training faster than a neural net without BN. However, adding BN not only changes the initialization, but also changes the whole training dynamics. I don't see how this paper separates the change of dynamics from initialization, and thus it is hard to interpret the meaning of "BN is a smart initialization".
3. I believe Benefit 2 is true in some sense, but the statement of Benefit 2 is not very clear. For providing a good theoretical explanation, the authors should at least provide the precise mathematical definitions of "margin" and "jitter", instead of just putting Figure 6 there. After that, the authors should either (1) use theoretical tools to bridge from jitter to large margin, and from large margin to generalization; or (2) measure margin, jitter, and generalization in experiments to justify their claim empirically.
4. The point that the stochasticity in batch statistics improves generalization is not new. See, e.g., Section 9.3 of [a recent survey by Huang](https://link.springer.com/content/pdf/10.1007/978-3-031-14595-7.pdf). I also recommend the authors go over all the related papers to better contextualize the current paper.

**Summary Of The Paper:**

This paper theoretically studies BN from the perspective of function approximation. The authors view the output function of a neural net as continuous piecewise affine splines. An important simplification in this paper is that it fixes the gamma and beta parameters in BN to 1 and 0 respectively. The key finding is that "BN is an unsupervised learning technique that adapts the geometry of a DN's spline partition to match the data". Then this paper points out two benefits of BN: (1) providing a "smart initialization" that "adapts weights to align the spline partition to the data"; (2) introducing a random jitter perturbation to the partition boundaries by varying BN statistics between mini-batches.

**Summary Of The Review:**

It is understandable that the current paper only makes intuitive claims because neural net with BN is a very complex object to study. However, I believe it is necessary to make the claims more formal so that the whole field can make progress. I feel that the current paper needs a major revision before getting accepted.

---

### Decision · Program_Chairs · 2023-01-20

**Decision:**

Reject

**Justification For Why Not Higher Score:**

Reviewer perspectives and scores clear; no author responses/revision.

**Justification For Why Not Lower Score:**

.

**Metareview: Summary, Strengths And Weaknesses:**

This is a nice paper providing a new explanation of batch norm.  Unfortunately, as identified by the reviewers, there are many limitations of the study, and the authors did not respond.  I urge the authors to continue their work and submit to a future venue.